# Pitch Control of Wind Turbine Blades Using Fractional Particle Swarm Optimization

**Ali Karami-Mollaee [1] and Oscar Barambones [2,*]** 

[1]   Faculty of Electrical and Computer Engineering, Hakim Sabzevari University, Sabzevar 9617976487, Iran
[2]   Automatic Control and System Engineering Department, University of the Basque Country, UPV/EHU, Nieves Cano 12, 48940 Vitoria, Spain
*   Correspondence: oscar.barambones@ehu.eus; Tel.: +34-945013235; Fax: +34-945013270

**Abstract:** To achieve the maximum power from wind in variable-speed regions of wind turbines (WTs), a suitable control signal should be applied to the pitch angle of the blades. However, the available uncertainty in the modeling of WTs complicates calculations of these signals. To cope with this problem, an optimal controller is suitable, such as particle swarm optimization (PSO). To improve the performance of the controller, fractional order PSO (FPSO) is proposed and implemented. In order to construct this approach for a two-mass WT, we propose a new state feedback, which was first applied to the turbine. The idea behind this state feedback was based on the Taylor series. Then, a linear model with uncertainty was obtained with a new input control signal. Thereafter, the conventional PSO (CPSO) and FPSO were used as optimal controllers for the resulting linear model. Finally, a comparison was performed between CPSO and FPSO and the fuzzy Takagi–Sugeno–Kang (TSK) inference system. The provided comparison demonstrates the advantages of the Taylor series with combination to these controllers. Notably, without the state feedback, CPSO, FPSO, and TSK fuzzy systems cannot stabilize WTs in tracking the desired trajectory.

**Keywords:** wind turbine; pitch angle control; fractional particle swarm optimization; fuzzy inference system; Taylor series

**MSC:** 93D15

## 1. Introduction

Solar or wind, as clean renewable viable energies, are accessible worldwide and are clean. However, due to economic reasons, the use of wind energy and wind turbines (WTs) is popular. There are two kinds of WT, fixed speed WTs (FWTs) [1,2] and variable-speed WTs (VWTs) [3]. It is not capable for FWTs to work such that the maximum power of wind can be harnessed [4]. Therefore, VWTs have recently been developed and constructed. To capture the maximum power of wind in VWTs, its operation regions are divided into four important sections using cut-out, rated, and cut-in boundaries [5]. Below the cut-out wind speed, VWTs will be shut down, to balance economic performance between the cut-out and rated wind speeds by controlling the generator torque [5]. Moreover, between the rated and cut-in boundaries of wind speeds, the pitch angle of turbine blades is used as the input control [6]. Finally, above the cut-in boundary, the VWT will be shut down again to protect it from fatigue damage [7].

On the other hand, mechanical stresses are another important challenge, which require powerful optimal or adaptive approaches to protect WTs [7]. Therefore, some pitch angle controllers have been proposed for WT blades between cut-out and rated wind speeds [6–14]. In [6], a digital controller was designed; classical controllers such as PID (proportional–integral–derivative) are proposed in [7,8]; a PID controller with an adaptive self-tuning regulator (STR) was constructed in [9]; a gain-scheduled PID controller was designed in [10]; a PI controller scheme is shown in [11]; and a combination of adaptive

and PI controllers is presented in [12]. Some simple nonlinear feedback controllers are proposed in [13,14]. Finally, to improve the performance of WTs, variable frequency converter controls to regulate the rotor speed were also used in [3,15].

Among these approaches, fractional controllers can have better performance [16,17] because they can precisely describe the behavior of many dynamical systems in physical, mathematical, and engineering fields [18–20]. Hence, many studies have focused on fractional subjects to develop their theories [21,22]. Therefore, the fractional calculations have progressed in various phenomena due to their applications in dynamic systems [17].

In the other hand, particle swarm optimization (PSO) is a power tool for optimization [23] and controllers [24]. Therefore, based on the advanced properties of fractional calculus and PSO, we improved the performance of conventional PSO (CPSO) using a combination of fractional and PSO. The proposed approach is fractional PSO (FPSO), and was applied to WTs for pitch angle control. Initially, state feedback was applied to the WT model; then, FPSO forced the WT rotor angular velocity to track its reference while the pitch angle of the blades was regulated. To demonstrate the advanced performance of FPSO, comparison was performed with CPSO and the Takagi–Sugeno–Kang (TSK) fuzzy system with similar parameters [25].

Hence, the proposed controller is demonstrated in five sections. First, the WT model and their subsystems are explained in Section 2. Then, the controller details, consisting of state feedback, with reference to rotor angular velocity, PSO, and TSK system, are provided in Section 3. The simulation results and comparisons of FPSO, CPSO, and TSK systems are presented in Section 4. Section 5 presents the conclusion.

## 2. Wind Turbine (WT) Model

The generator and drivetrain are two WT subsystems, their electrical and mechanical sections, respectively [16]. Other important subsystem of a WT is the aerodynamic section [16]. These subsystems are depicted in Figure 1.

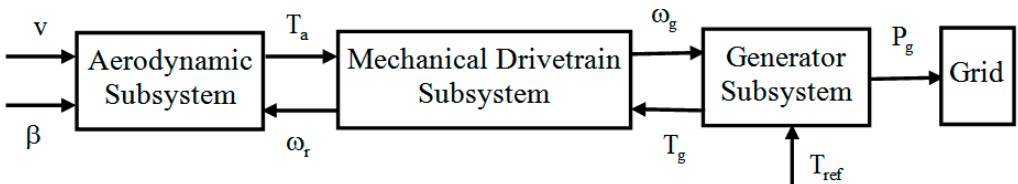

**Figure 1.** The WT subsystems.

### 2.1. The Aerodynamic Subsystem

Considering a WT with blade length r, power coefficient $C_p$, the captured power can calculated using the following equation:

$$P_a = \frac{\rho \pi r^2 v^2}{2} C_p(\beta, \lambda) \tag{1}$$

where $\rho$ is the air density and $v(t)$ is the wind speed, which are dependent on environment conditions. The power coefficient is dependent on the tip speed ratio, $\lambda$, and the pitch blades, $\beta$ [5], which are defined as follows:

$$\lambda = \frac{r\omega_r}{v} \tag{2}$$

$$C_p(\beta, \lambda) = \left( \frac{d_1 d_2}{\lambda_i} - d_1 d_3 \beta - d_1 d_4 \right) e^{-\frac{d_5}{\lambda_i}} + d_6 \lambda$$
$$\frac{1}{\lambda_i} = \frac{1}{\lambda + 0.08\beta} - \frac{0.035}{\beta^3 + 1} \tag{3}$$

where $\omega_r$ is the rotor side angular velocity of the turbine blades, and:

$$d_1 = 0.5176, d_2 = 116, d_3 = 0.4, d_4 = 5, d_5 = 21, d_6 = 0.0068 \tag{4}$$

Then, the generated rotor torque is described by:

$$T_a = \frac{P_a}{\omega_r} = \frac{\rho \pi r^3 v^2}{2\lambda} C_P(\beta, \lambda) \tag{5}$$

### 2.2. The Drivetrain Subsystem

The two-mass mechanical drivetrain, which shows the transient response and steady-state response in the presence of the controller, is described by the following equations and is depicted in Figure 2 [26].

$$\begin{aligned} J_r \dot{\omega}_r &= -K_r \omega_r + T_a - T_{ls} \\ J_g \dot{\omega}_g &= -K_g \omega_g - T_g + T_{hs} \end{aligned} \tag{6}$$

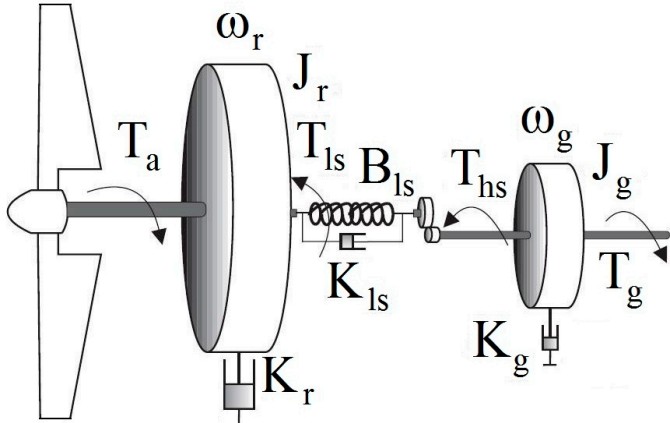

**Figure 2.** The drivetrain structure.

In Figure 2, $\omega_g$ and $\omega_r$ are angular velocity, $J_g$ and $J_r$ are inertia, $K_g$ and $K_r$ are the external dapping, $T_g$ and $T_a$ are the torque output, on the generator and rotor side, respectively, and finally, $T_{hs}$ and $T_{ls}$ are the braking torque in the high-speed and low-speed shaft. The ratio of gearbox is defined as:

$$n_g = \frac{\omega_g}{\omega_r} \tag{7}$$

Using the second part of Equation (6) results in:

$$J_g(n_g \dot{\omega}_r) = -K_g(n_g \omega_r) - T_g + \left(\frac{T_{ls}}{n_g}\right) \tag{8}$$

or:

$$n_g^2 J_g \dot{\omega}_r = -n_g^2 K_g \omega_r - n_g T_g + T_{ls} \tag{9}$$

Finally, adding this equation to the first part of Equation (6) results in [5]:

$$J_t \dot{\omega}_r = -K_t \omega_r + T_a - n_g T_g \tag{10}$$

Such that $J_t = J_r + n_g^2 J_g$ and $K_t = K_r + n_g^2 K_g$. In fact, all the parameters are transferred to a low-speed shaft [5].

### 2.3. The Generator Subsystem

The $T_g$, i.e., the output torque of the generator, can be modeled using the first-order dynamic, where $T_{ref}$ is the reference torque and $\tau_g = 15\,s$ is the generator time constant.

$$\dot{T}_g = \frac{T_{ref} - T_g}{\tau_g} \tag{11}$$

We focused on pitch control; thus, the generator torque reference was set as $T_{ref} = T_{reted}$. Moreover, the produced output power delivered to the grid can be written as $P_g = \eta_g \omega_g T_g$, where the efficiency of the generator is $\eta_g$.

### 3. The Optimal Controller Design

In this section, we first used state feedback and then calculated the desired rotor angular velocity. The CPSO and FPSO optimal controllers are also described.

### 3.1. State Feedback

Initially, we calculated the derivative of the power coefficient of Equation (3) with respect to the pitch angle of the blades.

$$\begin{aligned}\frac{dC_p}{d\beta} &= \frac{\partial C_p}{\partial \lambda_i}\frac{\partial \lambda_i}{\partial \beta} + \frac{\partial C_p}{\partial \beta} \\ &= \left(-\frac{c_1 c_2}{\lambda_i{}^2} + \frac{c_1 c_2 c_5}{\lambda_i{}^3} - \frac{c_1 c_3 c_5 \beta}{\lambda_i{}^2} - \frac{c_1 c_4 c_5}{\lambda_i{}^2}\right)\left(\frac{-0.08}{(\lambda + 0.08\beta)^2} + \frac{0.035}{(\beta^3 + 1)^2}\right)e^{-\frac{c_5}{\lambda_i}} - c_1 c_3 e^{-\frac{c_5}{\lambda_i}}\end{aligned} \tag{12}$$

Therefore, the Taylor series of Equation (5) around its optimal operating points $\beta_{opt}$ and $\lambda_{opt}$ would be:

$$T_a = \frac{\rho \pi r^3 v^2}{2\lambda_{opt}} \left.\frac{dC_p}{d\beta}\right|_{\substack{\beta = \beta_{opt} \\ \lambda = \lambda_{opt}}} (\beta - \beta_{opt}) + HOT \tag{13}$$

where HOT is used to denote higher-order terms; thus:

$$\dot{\omega}_r = -\frac{K_t}{J_t}\omega_r - \frac{n_g}{J_t}T_g + \frac{T_a}{J_t} = -\frac{K_t}{J_t}\omega_r - \frac{n_g}{J_t}T_g + \frac{\rho \pi r^3 v^2}{2J_t\lambda_{opt}} \left.\frac{dC_p}{d\beta}\right|_{\substack{\beta = \beta_{opt} \\ \lambda = \lambda_{opt}}} (\beta - \beta_{opt}) + \Delta \tag{14}$$

Due to the convergence of the Taylor series around the operating points, the unknown uncertainty $\Delta = \frac{HOT}{J_t}$ is bounded, i.e., $|\Delta| \le \eta$. Then, the following state feedback with the new input signal, u, and the arbitrary parameter, a, can be used.

$$\beta = \frac{\left(\frac{n_g}{J_t}\right)T_g + \left(\frac{K_t}{J_t} - a\right)\omega_r + u}{\frac{\rho \pi r^3 v^2}{2J_t\lambda_{opt}} \left.\frac{dC_p}{d\beta}\right|_{\substack{\beta = \beta_{opt} \\ \lambda = \lambda_{opt}}}} + \beta_{opt} \tag{15}$$

Then, system Equation (14) can be rewritten as follows:

$$\dot{\omega}_r = -a\omega_r + u + \Delta \tag{16}$$

In which $\Delta$ is an unknown uncertain function. We aimed to design an optimal approach such that in this linear system, the rotor angular velocity, $\omega_r$, tracked the desired signal, $\omega_{rd}$.

### 3.2. Reference of Rotor Angular Velocity

As mentioned in the Introduction and based on Figure 3, the VWT operation modes were divided into four regions using wind speed boundaries of cut-in, rated, and cut-out. The critical point is the rated wind speed, such that below this point, the pitch of turbine

blades is fixed and generator torque is controlled; hence, the rotor speed is increased to have the maximum of power coefficient. Moreover, above the rated wind speed, the generator reference torque is fixed and is set to its rated value. In this region, the pitch angle would be increased to reduce the rotor speed. Finally, out of the cut-in and cut-out wind speeds, the turbine would be shut down due to the economic criterion and fatigue damages, respectively [5].

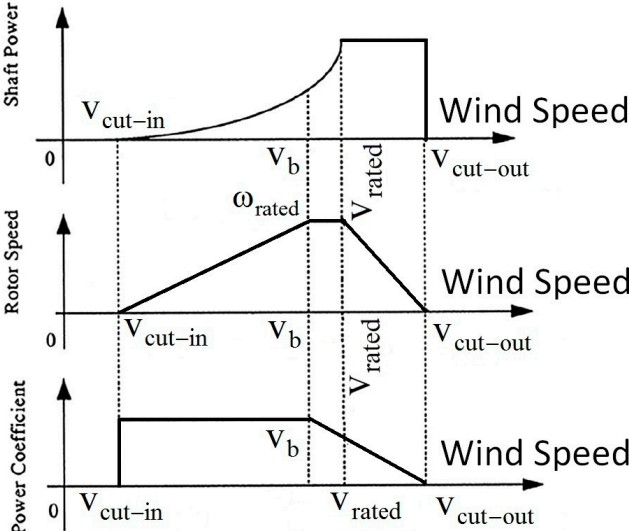

**Figure 3.** Operation regions of the VWT.

In this study, we focused on the pitch angle control in region three, whereas the rotor angular velocity should be reduced with the increased of wind speed. Therefore, the reference of rotor angular velocity is as follows:

$$\omega_{rd} = \omega_{rated} - \omega_{rated} \frac{v - v_{rated}}{v_{cut-out} - v_{rated}} \tag{17}$$

Based on Equation (2), one can conclude that:

$$\omega_{rated} = \frac{\lambda_{opt} v_{rated}}{r} \tag{18}$$

### 3.3. Particle Swarm Optimization (PSO) and Controller Structure

According to the previous sections, the aim was to determine the angular velocity of the rotor, i.e., $\omega_r$ tracks the desired trajectory, $\omega_{rd}$. To this end, the error, $e = (\omega_r - \omega_{rd})^2$, is applied to the PSO. PSO is applied to calculate the input control signal, u, in Equation (16), while the error signal, e, converges to zero. In CPSO, the velocity of each particle is updated as follows [23]:

$$\dot{v}_i(t) = c_1 \, \phi_1 \, (p_b - x_i(t)) + c_2 \, \phi_2 \left( p_g - x_i(t) \right) : i = 1, 2, \dots, n \tag{19}$$

where n is the number of particles, $v_i(t)$ is the velocity of each particle, $\phi_1$ and $\phi_2$ are uniformly random functions between 0 and 1, $p_b$ is the best position of each particle, $p_g$ is the global best position between all of the particles, $x_i(t)$ is the current position of the each particle, and coefficients $c_1$ and $c_2$ are constant numbers. Then, the position of any particle is updated as follows [23]:

$$\dot{x}_i = v_i(t) \tag{20}$$

There are several definitions for fractional differentiations and integrations, such as Grünwald–Letnikov, Riemann–Liouville, and Caputo formulae [27]. Among them, the

Caputo method is popular because initial conditions are considered [17,27]; thus, the Caputo definition was used in this study.

**Definition 1.** *Caputo q-order integration and the differentiation of variable* $v(t)$ *with respect to time,* $t$, *is defined as follows* [27]:

$$
{}_{t_0}I_t^q v(t) = \frac{1}{\Gamma(q)} \int_{t_0}^t (t-\tau)^{q-1} v(\tau) d\tau \tag{21}
$$

$$
{}_{t_0}D_t^q v(t) = \frac{1}{\Gamma(1-q)} \int_{t_0}^t \frac{v'(\tau)}{(t-\tau)^q} d\tau \tag{22}
$$

*Here,* $t > t_0$ *and* $t_0$ *is the initial time;* $0 < q < 1$ *and* $\Gamma(q) = \int_0^\infty \tau^{q-1} e^{-\tau} d\tau$ *is the Gamma function.*

**Remark 1.** *In this study, we considered the zero initial condition, i.e.,* $t_0 = 0$. *Moreover, for simplicity, subscript* $t$ *was also eliminated; hence, we use* $D^q v(t)$ *instead of* ${}_{t_0}D_t^q v(t)$ *and* $I^q v(t)$ *instead of* ${}_{t_0}I_t^q v(t)$.

To improve the performance of the CPSO, we propose FPSO, as follows:

$$
D^q v_i(t) = c_1 \phi_1 (p_b - x_i(t)) + c_2 \phi_2 \left(p_g - x_i(t)\right) : i = 1, 2, \ldots, n \tag{23}
$$

For a valid comparison, both CPSO and FPSO were implemented. Therefore, the implemented diagram of the proposed approach is illustrated in Figure 4. From this figure, one can see the combination of two feedbacks for nonlinear systems of Figures 1 and 2. The first state feedback of Equation (15) is based on the theory of the Taylor series with a new input control signal, $u(t)$, to obtain a linear system, as in Equation (16). Then, in the second feedback, FPSO or CPSO were applied to this linear system in order to minimize the error signal. Moreover, we used the fuzzy TSK systems.

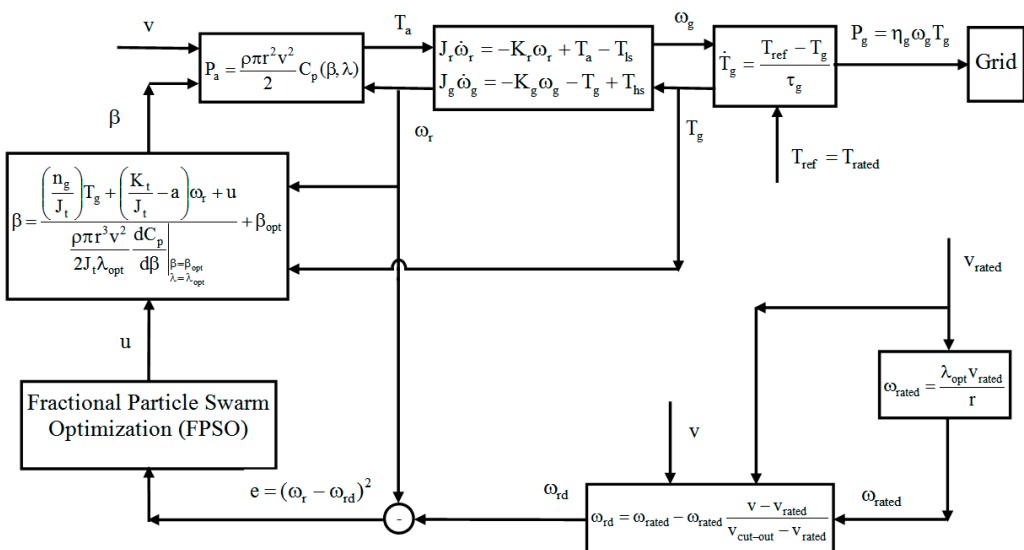

**Figure 4.** The implemented structure of the proposed controller.

### 3.4. Takagi-Sugeno-Kang (TSK) Controller Structure

The structure of proposed TSK controller is shown in Figure 4, with two inputs $e = (\omega_r - \omega_{rd})$ and its derivative and one output $u(t)$. For each input and output, five triangular membership functions are defined as negative-large (NL), negative-small (NS), zero (Z), positive-small (PS), and positive-large (PL). The range of inputs was set between

−2 and +2, and the range of output was also set to between −200 and 200. Therefore, 25 rules with the aggregation defuzzification were used.

## 4. Simulations Results

We used the two-mass 5 MW, VWT in National Renewable Energy Laboratory (NREL) located at Colorado, with the aerodynamic parameters in Table 1 and drivetrain parameters in Table 2 [28].

**Table 1.** The aerodynamic parameters of VWT.

| Parameter | Value | Unit |
|---|---|---|
| $T_{rated}$ | $4.3094 \times 10^4$ | $N \times m$ |
| $v_{cut-in}$ | 3 | m/s |
| $v_{rated}$ | 10.5 | m/s |
| $v_{cut-out}$ | 25 | m/s |
| $\beta_{opt}$ | 0 | deg |
| $\lambda_{opt}$ | Scalar | 7.55 |

**Table 2.** The mechanical drivetrain parameters of VWTs.

| Notation | Value | Unit |
|---|---|---|
| r | 21.62 | m |
| $\rho$ | 1.308 | $kg/m^3$ |
| $J_r$ | $3.25 \times 10^5$ | $kg \times m^2$ |
| $J_g$ | 34.4 | $kg \times m^2$ |
| $K_r$ | 27.36 | $(N \times m)/(rad/s)$ |
| $K_g$ | 0.2 | $(N \times m)/(rad/s)$ |
| $K_{ls}$ | $9.5 \times 10^3$ | $(N \times m)/rad$ |
| $B_{ls}$ | $2.691 \times 10^5$ | $(N \times m)/(rad/s)$ |
| $n_g$ | 43.165 | Scalar |

For a reliable comparison, all of the simulations were performed using MATLAB software with a sample time of 0.01. The wind speed is shown in Figure 5, with a mean value 16 and maximum disturbance of 5. Notably, this is between 11.4 and 25, i.e., in region 3: $v_{rated} = 11.4 < v(t) < v_{cut-out} = 25$. In addition, the initial value for the rotor angular velocity is set to 3, i.e., $\omega_r(0) = 3$, and the feedback parameter is set as a = 2. Moreover, Figure 6 shows the reference of rotor speed in region 3 denoted by Equation (17).

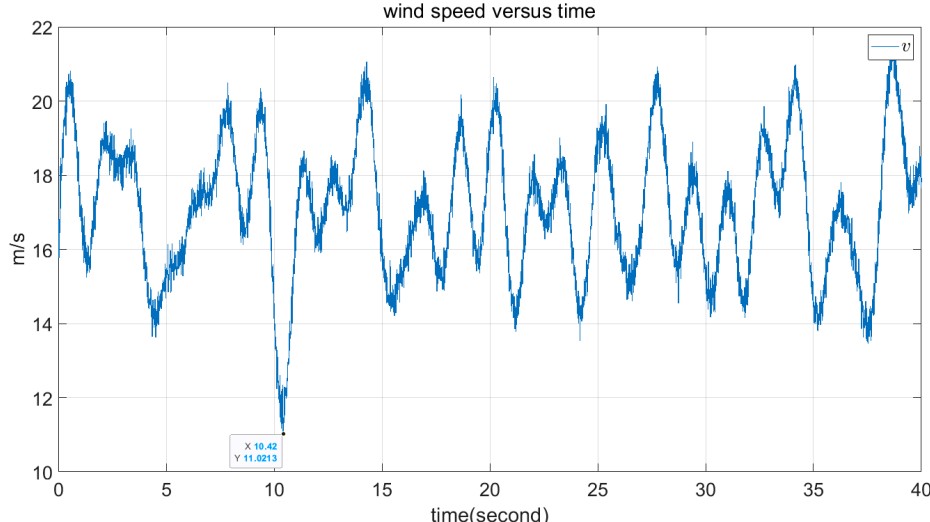

**Figure 5.** The profile of time series wind speed.

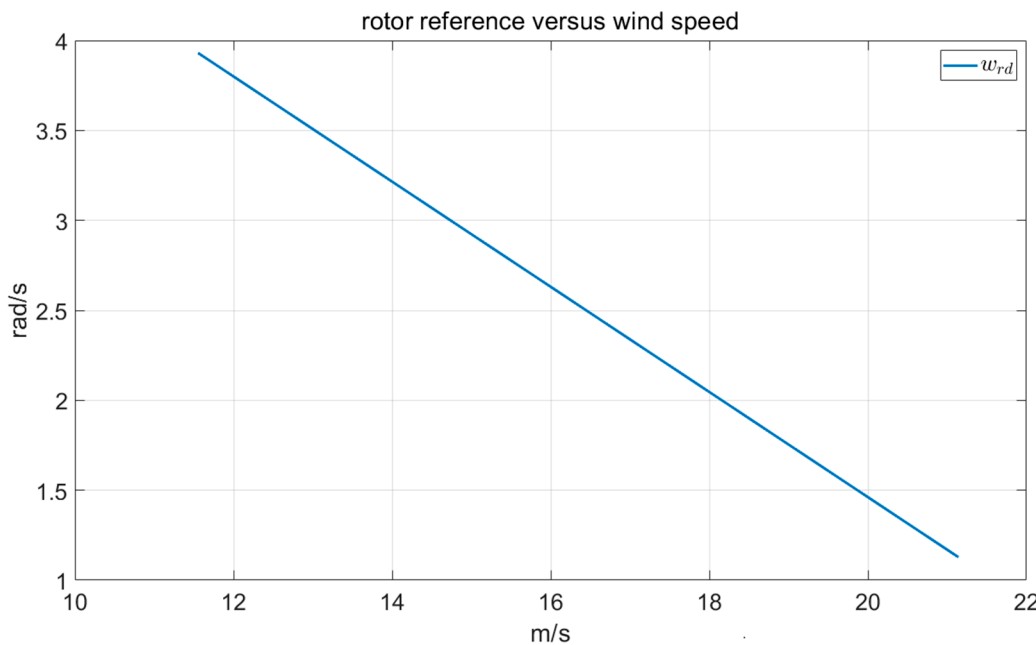

**Figure 6.** Reference of the rotor speed in region 3 of VWTs.

**Example 1.** *The CPSO approach*.

As the first result, simulations of the CPSO in Equation (19) are shown in Figures 7–10. The parameters of the PSO are $c_1 = c_2 = 1.5$, with 20 particles and 4000 iterations.

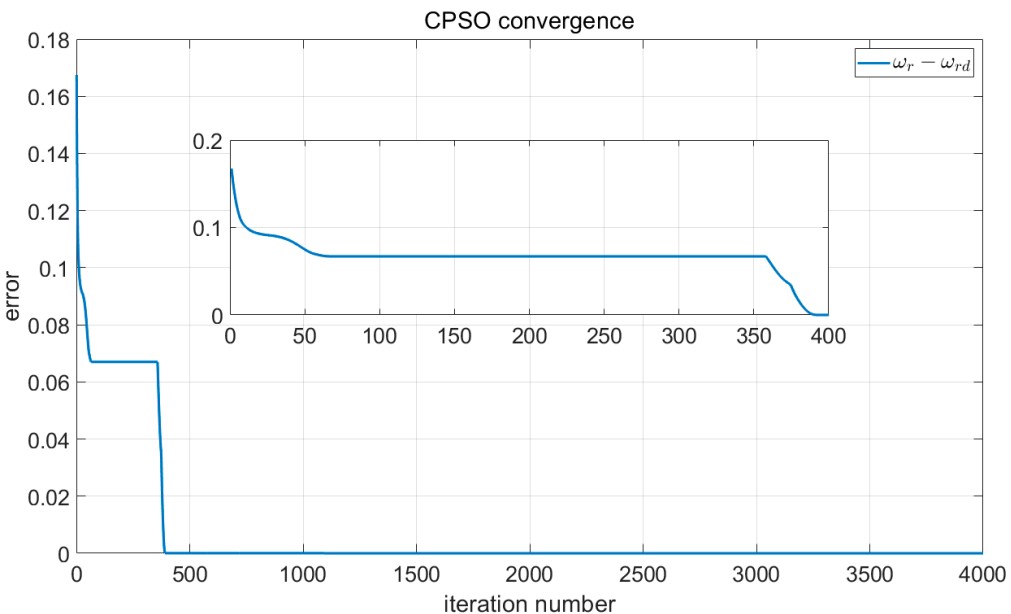

**Figure 7.** Convergence of the CPSO.

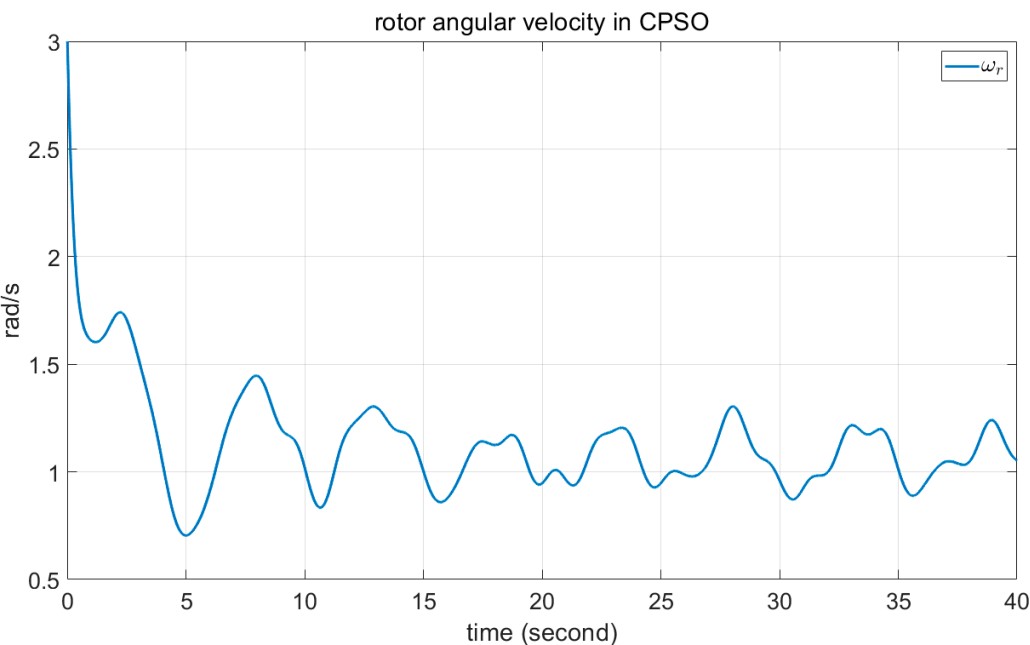

**Figure 8.** Angular velocity of the rotor in CPSO.

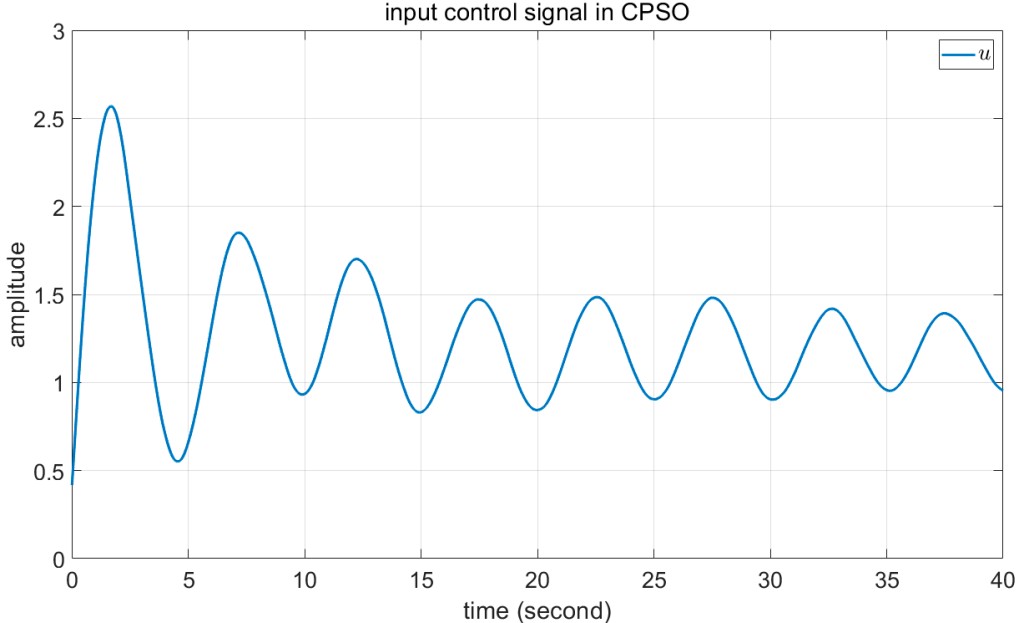

**Figure 9.** The input control signal of state feedback in CPSO.

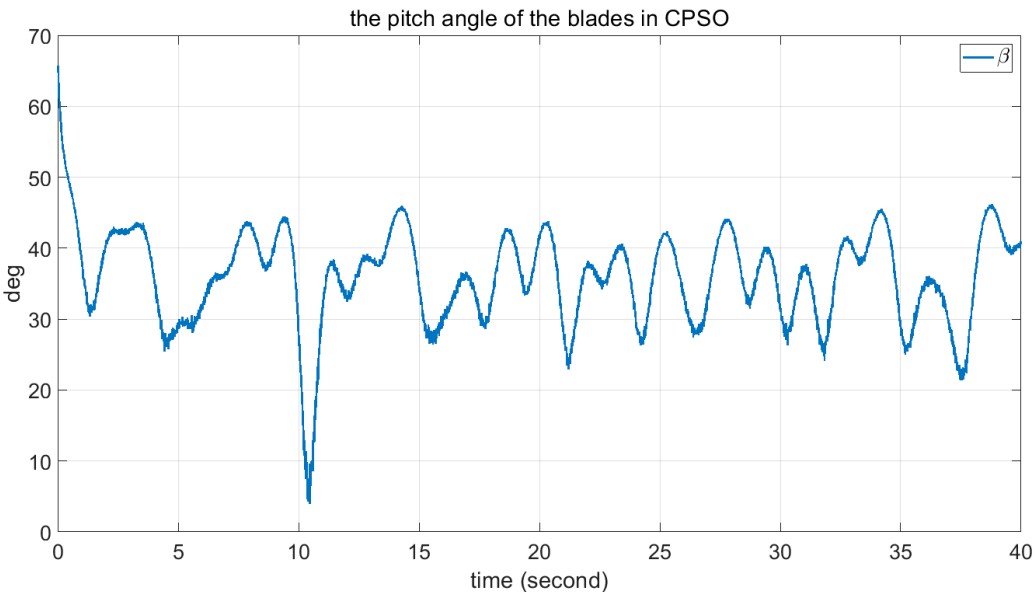

**Figure 10.** The angle of blades in CPSO.

**Example 2.** *The proposed FPSO approach*.

As the second result, simulation of the FPSO in Equation (21) is shown in Figures 11–14 with the fraction order q = 0.8 and the same parameters of example 1, i.e., with 20 particles and 4000 iterations and $c_1 = c_2 = 1.5$.

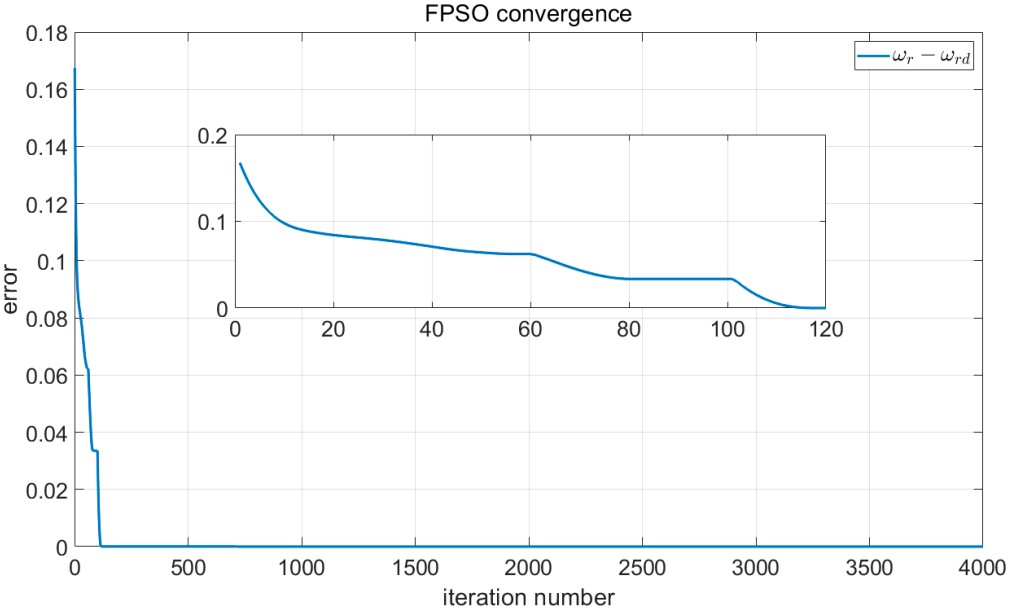

**Figure 11.** Convergence of the FPSO.

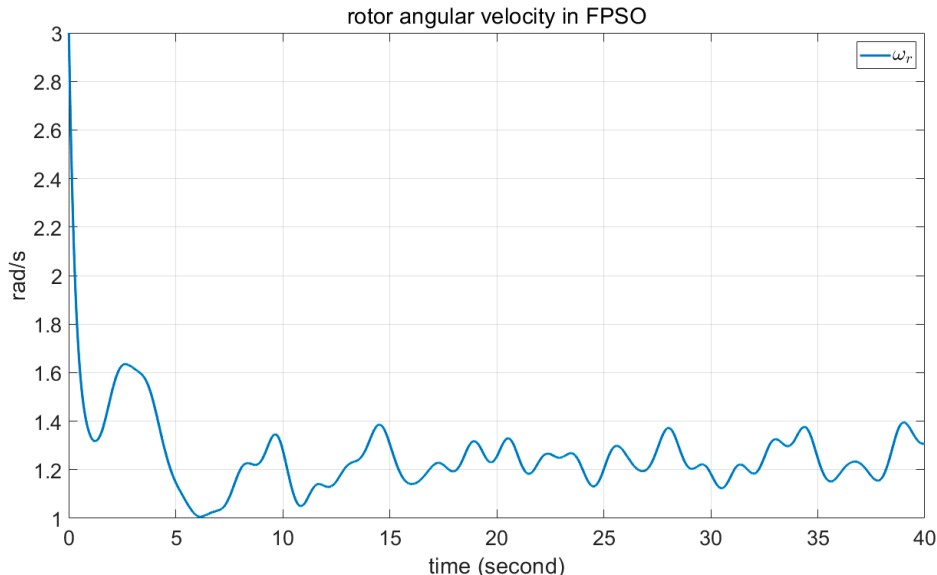

**Figure 12.** Angular velocity of the rotor in FPSO.

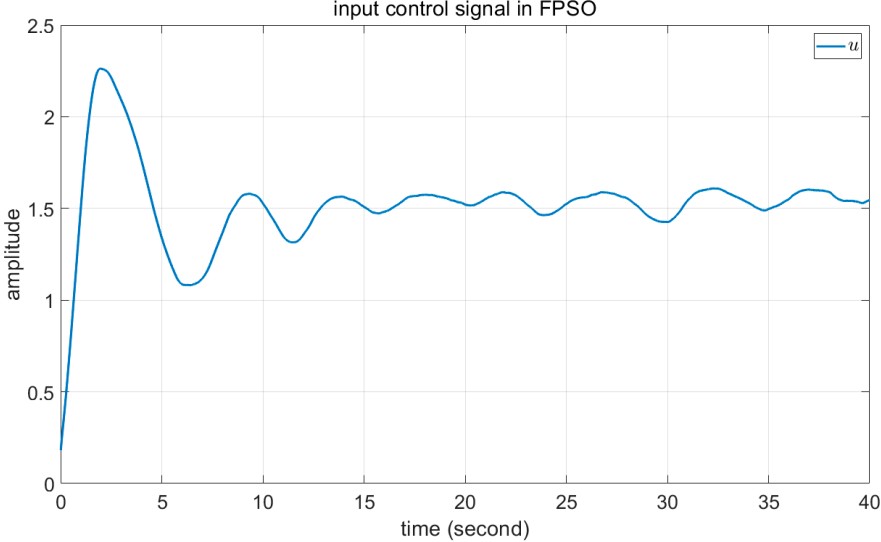

**Figure 13.** The input control signal of state feedback in FPSO.

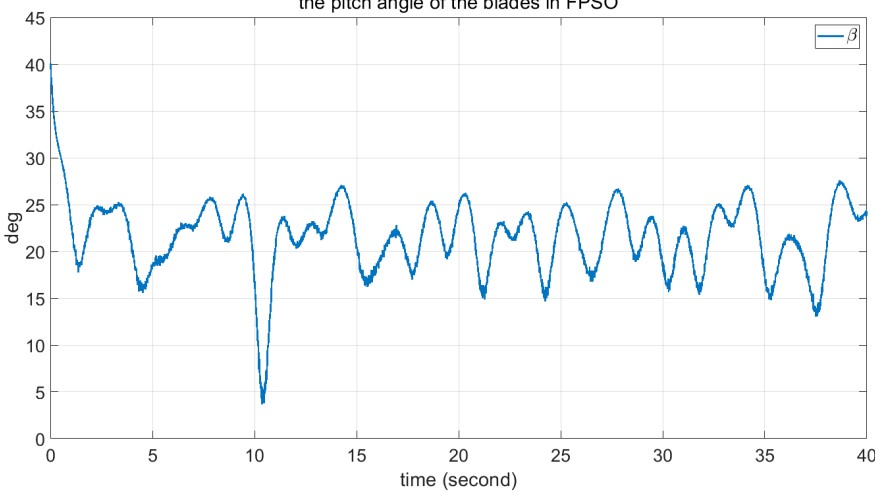

**Figure 14.** The angle of blades in FPSO.

**Example 3.** *The proposed fuzzy TSK approach.*

As the third example, the results of the fuzzy TSK inference system are shown in Figures 15–18, similar to previous examples.

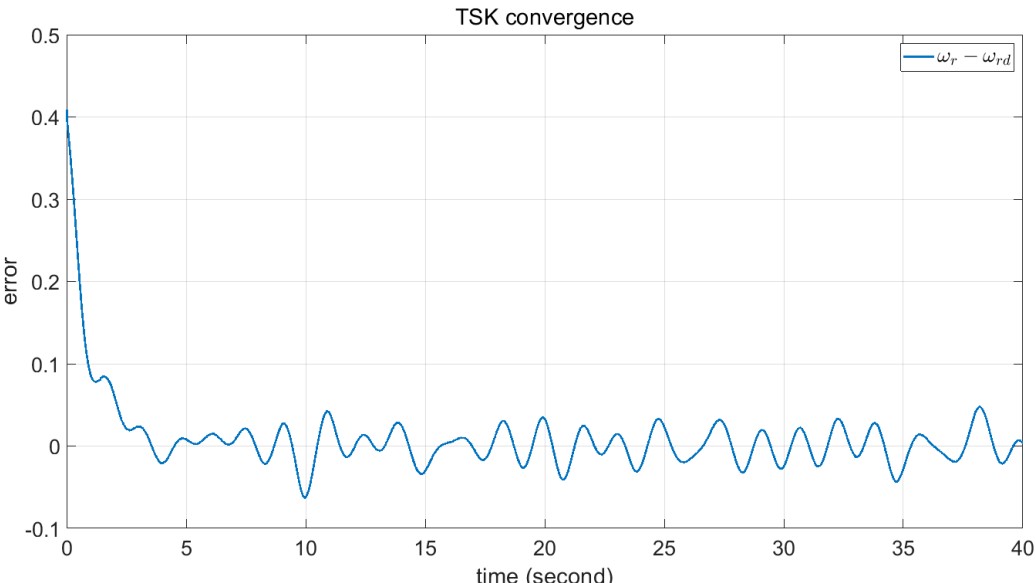

**Figure 15.** Convergence of the fuzzy TSK.

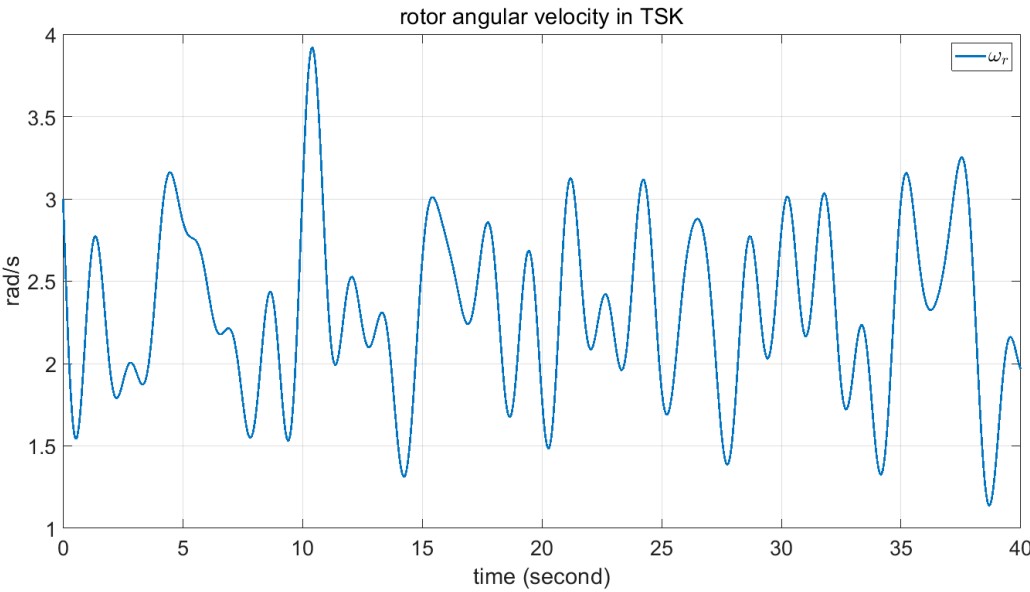

**Figure 16.** Angular velocity of the rotor in fuzzy TSK.

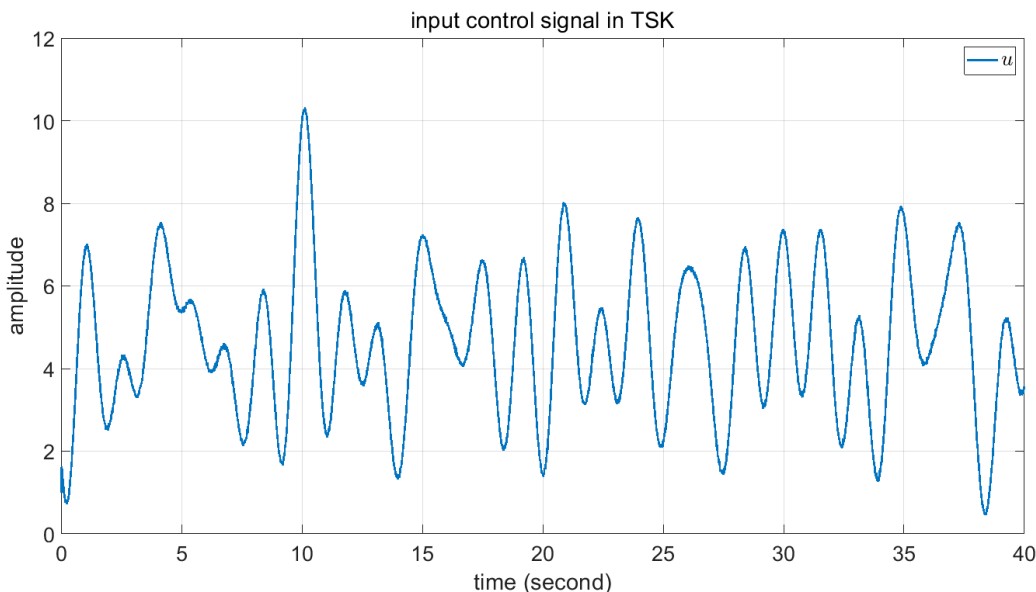

**Figure 17.** The input control signal of state feedback in fuzzy TSK.

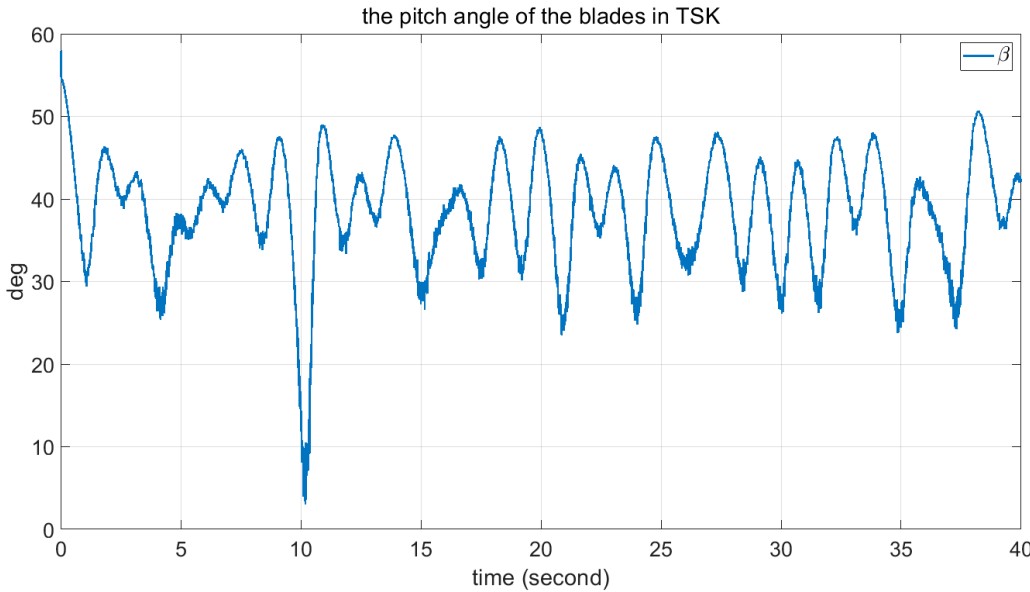

**Figure 18.** The angle of blades in fuzzy TSK.

**Comparison Results.** FPSO, CPSO, and proposed TSK approach.

Comparisons of these figures demonstrate the good performance of the proposed FPSO. Figures 7 and 11 show the convergence of the error signal. These figures indicate the faster convergence of FPSO. However, as shown in Figure 15, the convergence error of TSK seems to be very good. Comparison of Figures 8, 12 and 16 shows the smaller variation in rotor angular velocity in FPSO. Moreover, as Figures 9, 13 and 17 show, the input control signal of the state feedback in FPSO is smoother. In addition, Figures 10, 14 and 18 show the pitch angle of WT blades. The variation in pitch in FPSO is smaller in comparison with the two other approaches. These are cause low mechanical stress to the drivetrain part in FPSO. Thereafter, when the wind speed is decreased to about 10.4 s, the pitch angles are converged to their optimal value, $\beta_{\text{opt}} = 0$. Notably, when the wind speed is decreased below $v_{\text{rated}} = 11.4$, the pitch angles are fixed to the optimal value and the WT torque would be controlled. Finally, in Figures 19 and 20, the time for one rotation and its mean value is also depicted. From Figure 19, one can see the low speed variation in FPSO

and very high speed variation in TSK controller. Figure 20 shows the closeness of FPSO and CPSO for one rotation, but the fuzzy TSK system has very low speed. As the final result, one can see the simplicity and good performance of FPSO in concept and real-world implementation with respect to the CPSO and TSK inference systems.

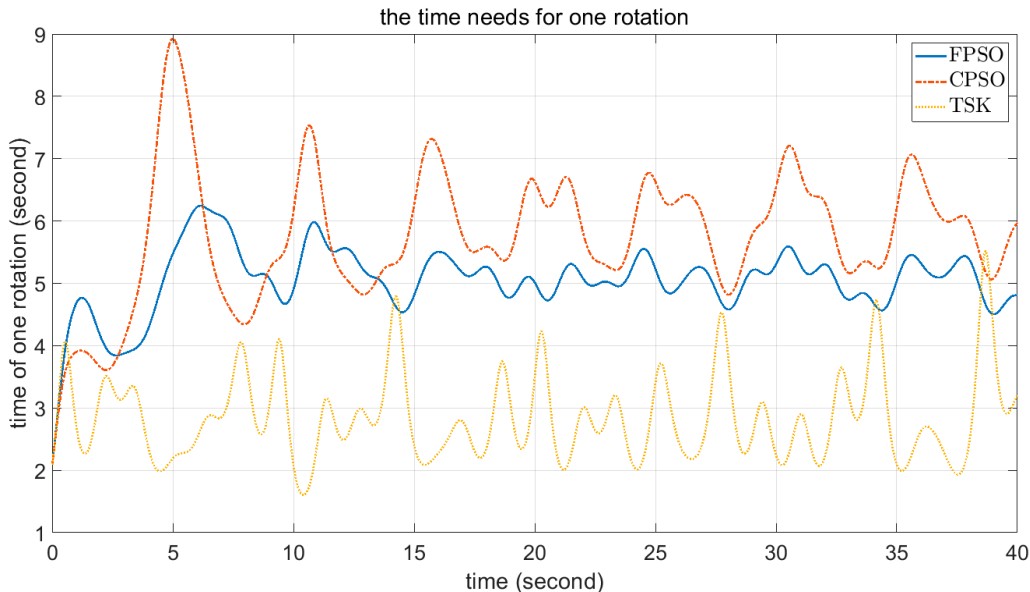

**Figure 19.** The time for one rotation in FPSO, CPSO, and fuzzy TSK.

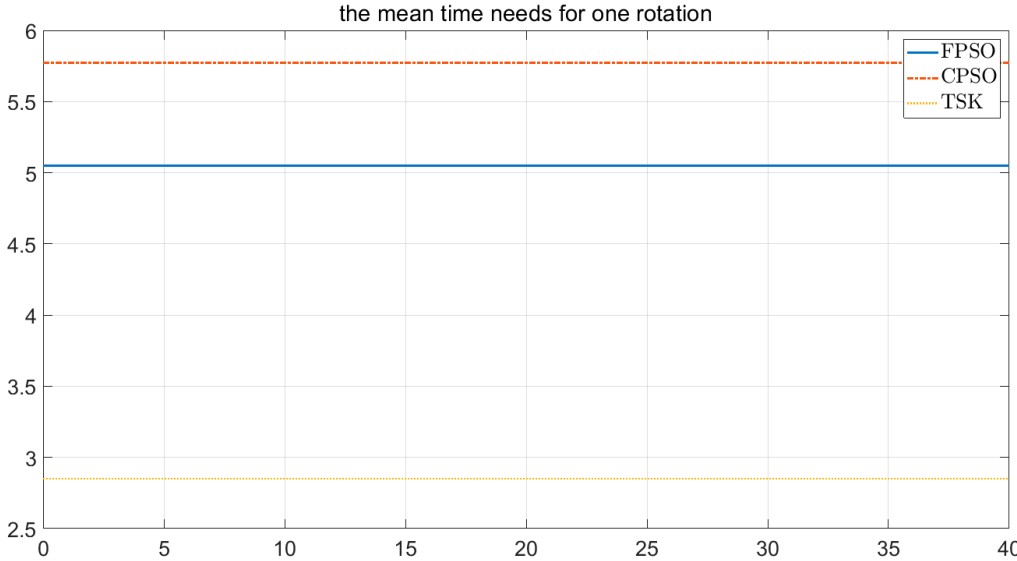

**Figure 20.** The mean time for one rotation in FPSO, CPSO, and fuzzy TSK.

**Remark 2.** *Notably, unlike nonlinear systems, in linear systems only one global minimum is available for the error* $\mathrm{e} = (\omega_{\mathrm{r}} - \omega_{\mathrm{rd}})$. *Therefore, in the absence of the first feedback, the FPSO, CPSO, and TSK cannot converge.*

## 5. Conclusions

In this paper, new issues are presented for the optimal pitch control of variable-speed wind turbines (VWTs). First, a new state feedback approach based on Taylor series was applied to the wind turbine (WT) to obtain a linear model with uncertainty and a new input control signal. Second, particle swarm optimization (PSO) and the fuzzy Takagi–Sugeno–

Kang (TSK) system were used to force the rotor angular velocity to track the desired trajectory. Then, the overall closed loop approach was applied to the two-mass 5 MW WT. Finally, comparisons of the conventional PSO (CPSO), fractional PSO (FPSO), and TSK controller were performed using simulation. In the proposed FPSO, the performance of the controller was better. To show the superiority of the FPSO, the same parameters were used in all three approaches. Moreover, the design procedures show the simplicity of the FPSO in concept and in realization

**Author Contributions:** A.K.-M.: analysis, writing—original draft preparation; O.B.: conceptualization, writing—review and editing. All authors have read and agreed to the published version of the manuscript.

**Funding:** This study received no external funding.

**Institutional Review Board Statement:** Not applicable.

**Informed Consent Statement:** Not applicable.

**Data Availability Statement:** Not applicable.

**Acknowledgments:** The authors wish to express their gratitude to the Basque Government, through the project EKOHEGAZ (ELKARTEK KK-2021/00092), to the Diputación Foral de Álava (DFA), through the project CONAVANTER, and to the UPV/EHU, through the project GIU20/063, for supporting this study.

**Conflicts of Interest:** The authors declare no conflict of interest.

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
