# Peer review of "Pitch Control of Wind Turbine Blades Using Fractional Particle Swarm Optimization"

_axioms, doi:10.3390/axioms12010025_

Round 1

Reviewer 1 Report

Dear Authors,

In my opinion in real wind turbine rotor speed changes more slowly due to big inertia. Presented results are very theoretical.

Author Response

Response to Reviewers

Manuscript ID: axioms-2077800

On Control of Pitch Blades Wind Turbine Using Fractional Particle Swarm Optimization

Authors: Ali Karami-Mollaee and Oscar Barambones

First of all, the authors would like to thank the associate editor and the respected reviewers, who gave us many constructive comments and valuable suggestions in order to improve this paper. The authors have revised the paper according to the reviewers’ comments. The responses to the reviewer comments can be found below their respective comments and the changes made in the paper are marked in red color.

Moreover, based on the comments of editor and reviewer, references 1, 2, 3, 4, 9, 24 and 30 are removed. Other references are rearranged accordingly.

Sincerely Yours,

The corresponding author,

Reviewers' comments and our responses:

Reviewer #1:

COMMENTS FOR THE AUTHOR:

  1. In my opinion in real wind turbine rotor speed changes more slowly due to big inertia. Presented results are very theoretical.

Response to the reviewer:

  • Thanks to this reviewer’s comment, it is true that this comment could be true for some big wind turbines but not for other smaller ones, which have less mechanical inertia. Nevertheless, according to the figure 8, in conventional PSO (CPSO) the variations of rotor speed is about 0.4 radian per second. Moreover, based on the figure 12, in fractional PSO (FPSO) the variations of rotor speed is about 0.3 radian per second, which are not a disproportionate values. Moreover, it should be noted that these big changes are due to the big changes introduced in the wind speed in order to show the performance of the proposed controller.

Reviewer 2 Report

interesting work, the work discusses the pitch angle control of a wind turbine in order to produce the maximum of electrical energy and also to avoid stopping the wind turbine in the case of strong wind speeds, I recommend to the author of:

- to further explain the problem and the proposed solution

- the idea is to regulate the pitch angle, to improve the performance of the wind turbine, but we must discuss in the introduction the other solutions to improve it, which parts of the system must be regulated, not just the pitch angle.

- the regulation system is simple for the pitch, in my opinion it would be better to add at least the MPPT command to control the speed and discuss the control techniques for the converters and the machine

- the simulation results are interesting, but they must be improved to enhance this work and also a comparison is mandatory with other works

Author Response

Response to Reviewers

Manuscript ID: axioms-2077800

On Control of Pitch Blades Wind Turbine Using Fractional Particle Swarm Optimization

Authors: Ali Karami-Mollaee and Oscar Barambones

First of all, the authors would like to thank the associate editor and the respected reviewers, who gave us many constructive comments and valuable suggestions in order to improve this paper. The authors have revised the paper according to the reviewers’ comments. The responses to the reviewer comments can be found below their respective comments and the changes made in the paper are marked in red color.

Moreover, based on the comments of editor and reviewer, references 1, 2, 3, 4, 9, 24 and 30 are removed. Other references are rearranged accordingly.

Sincerely Yours,

The corresponding author,

Reviewers' comments and our responses:

Reviewer #2:

Interesting work, the work discusses the pitch angle control of a wind turbine in order to produce the maximum of electrical energy and also to avoid stopping the wind turbine in the case of strong wind speeds, I recommend to the author of:

COMMENTS FOR THE AUTHOR:

  1. To further explain the problem and the proposed solution.

Response to the reviewer:

  • Thanks to this reviewer’s comment, to fulfill this comment we add further explain about the proposed approaches in the end of simulation section.
  1. The idea is to regulate the pitch angle, to improve the performance of the wind turbine, but we must discuss in the introduction the other solutions to improve it, which parts of the system must be regulated, not just the pitch angle.

Response to the reviewer:

  • I section 3.2 it is completely explained the operation modes of variable speed wind turbine (VWT). We explain that there is a critical wind speed known as rated, such that below that pitch angle of WT blades is fixed and generator torque is controlled, hence the rotor speed is increasingly to have the maximum of power coefficient. Moreover, above the rated wind speed the generator reference torque is fixed and is set to its rated value. In this region, the pitch angle would be increased to reduce the rotor speed. In this case the blades angle is controlled.
  • Anyway, the following text with references 15 and 16 are added in introduction: “Finally, to improve the performance on the WT, the variable frequency converter control to regulate the rotor speed is also used in [15,16]”.
  1. The regulation system is simple for the pitch, in my opinion it would be better to add at least the MPPT command to control the speed and discuss the control techniques for the converters and the machine.

Response to the reviewer:

  • Unfortunately we do not understand this comment because, as it is indicated in section 3.2, in this paper we have focused on pith angle control in region three and in this region the reference for the rotor speed is given by Eq. (17). It is true that for other regions, the reference for the rotor speed could be given by an MPPT, but in this case the pitch angle should be fixed at its optimal value in order to maximize the generated power. It is true that in the literature, one can find several control techniques for the converters and the electrical generator of the wind turbines but this is out of the scope of this work (Please refer to references 15 and 16 in the paper).
  1. The simulation results are interesting, but they must be improved to enhance this work and also a comparison is mandatory with other works.

Response to the reviewer:

  • Thanks to this reviewer’s comment, to fulfill this comment the Takagi-Sugeno-Kang (TSK) fuzzy inference system is simulated. Comparison is also done in detailed.

Round 2

Reviewer 2 Report

all comments have been well answered, it's ok for me